# Linear Regression vs. Deep Learning: A Simple Yet Effective Baseline for Human Body Measurement

**DOI:** 10.3390/s22051885

**Published:** 2022-02-28

**Authors:** Kristijan Bartol, David Bojanić, Tomislav Petković, Stanislav Peharec, Tomislav Pribanić

**Affiliations:** 1Faculty of Electrical Engineering and Computing, University of Zagreb, Unska 3, 10000 Zagreb, Croatia; david.bojanic@fer.hr (D.B.); tomislav.petkovic.jr@fer.hr (T.P.); tomislav.pribanic@fer.hr (T.P.); 2Peharec Polyclinic for Physical Medicine and Rehabilitation, 52100 Pula, Croatia; peharec@peharec.com

**Keywords:** body measurement, linear regression, statistical models, anthropometry, SMPL, shape estimation, mesh regression, virtual try-on

## Abstract

We propose a linear regression model for the estimation of human body measurements. The input to the model only consists of the information that a person can self-estimate, such as height and weight. We evaluate our model against the state-of-the-art approaches for body measurement from point clouds and images, demonstrate the comparable performance with the best methods, and even outperform several deep learning models on public datasets. The simplicity of the proposed regression model makes it perfectly suitable as a baseline in addition to the convenience for applications such as the virtual try-on. To improve the repeatability of the results of our baseline and the competing methods, we provide guidelines toward standardized body measurement estimation.

## 1. Introduction

Body measurement or anthropometry is a study of the numerical description of human body segments and overall shape. Anthropometry is important for health applications [1], virtual try-on and the fashion industry [2], fitness [3], and ergonomics [4]. Advances in human body measurement and shape estimation have been significantly driven by statistical models and deep learning. The statistical models such as SMPL [5] describe the population of human bodies using pose and shape parameters. The parameters correspond to specifically posed and shaped template meshes. The template meshes are particularly useful for body measurement because each mesh contains a fixed number of vertices, and their semantics are common for the entire population of meshes. Therefore, the procedure for body measurement from the template meshes for a given statistical model, e.g., SMPL, can be standardized. We describe the procedure for extracting 15 body measurements from SMPL templates in Section 3 and use these measurements to compare the linear regression model to the state-of-the-art.

Statistical models enable human body mesh regression, which can be described as fitting point cloud (3D) or image (2D) features to the pose and shape parameters of a given template mesh of the statistical model to obtain the best correspondence between the estimated mesh and the input human body. The 3D point cloud-based approaches [6,7,8] are generally more accurate than their image-based counterparts [9], but they require 3D scans as input, making them impractical for most anthropometric applications. Image-based methods greatly simplify the data acquisition stage. A very popular research topic is pose and shape estimation using in-the-wild images of clothed people in various difficult poses [10,11,12,13,14,15,16,17,18,19,20]. Even though these models recover a person’s pose remarkably well, less attention is dedicated to the accurate estimation of shape parameters. Finally, there are several image-based methods that estimate the shape parameters based on the extracted silhouette(s) of a person [21,22,23,24,25,26,27,28,29]. Silhouette-based methods achieve state-of-the-art results in shape estimation and body measurement, but their applications are limited to a configuration with minimal clothing and fixed posture.

We propose a linear regression model that requires only the information that any person can self-estimate, such as height and weight. We demonstrate strong performance of the proposed model against the state-of-the-art for body measurement estimation on two public datasets, BODY-fit [28] and ANSUR [30]. The linear model provides a simple and straightforward way to obtain an estimate of body measurements, which makes it convenient for virtual try-on and augmented reality. In this work, we focus on measurement estimation of the bodies from the statistical human body population. Statistical models such as SMPL are extensively used today, especially in computer vision, so we propose the linear model as a baseline for future evaluations and provide guidelines for body measurement from template meshes.

The main contributions of our work are the following:We propose the linear regression model, that uses self-estimated height and weight, to be used as a baseline for human body measurement estimation;We demonstrate that the baseline performs strongly against the state-of-the-art methods for body measurement estimation and analyze its performance in detail;We publish the source code, the demo for obtaining the body measurements given height and weight as input, and the protocol for extracting the standard body measurements from the SMPL mesh.

## 2. Related Work

Body measurements can be obtained using traditional anthropometric techniques and tools, 3D scanning data such as point clouds or meshes (non-parametric approaches), or features extracted from 3D or 2D data and then used to estimate the parameters of the statistical model (parametric approaches). The accuracy of body measurements obtained using traditional tools and 3D scanning are usually more accurate compared to feature-based, parametric approaches, but the former are more time consuming and relatively expensive.

**Traditional Anthropometry.** Traditional body measurement involves the use of tools such as calipers and tape measures [2]. Measurements taken by an expert are considered the gold standard and are generally used as the ground truth [31]. The public anthropometric databases such as ANSUR are collected by the expert measurers. However, even measurements taken by the experts are not perfectly accurate [32]. Therefore, the values for the allowable or “expert” error [30] for each measurement are defined in the standard ISO:7250 [33,34]. Based on the public databases, several works propose linear regression models for the estimation of measurements [35,36,37] and other body characteristics such as skeletal muscle mass [38]. We extend these analyses by focusing specifically on using self-estimated height and weight as input as well as on using the statistical body models.

**Non-Parametric Approaches.** With the advances in 3D scanning technology, more automatic approaches to body measurement have been proposed [7,39,40]. Most 3D-based body measurement methods use landmarks to determine distances and calculate measurements such as arm and leg length, shoulder-to-crotch, etc. Circumferences can be obtained by slicing the point cloud with a vertical plane and summing up distances between the nearest points [39]. The 3D-based methods are generally the most accurate among the (semi-)automatic body measurement methods. However, their main drawback is that they require 3D scanning, which is cumbersome and relatively expensive.

There are a number of image-based (2D) non-parametric models [20,41,42,43] that freely deform meshes to best fit the input features. To improve convergence, they start with the template human mesh. However, the final deformed mesh does not necessarily retain the original vertex semantics. This property makes current image-based non-parametric models less suitable for body measurements. On the other hand, compared to their parametric counterparts, non-parametric models might have the advantage of better fitting out-of-distribution samples.

**Parametric Approaches.** Apart from direct body measurements from point clouds, 3D scans are usually used to create the statistical models [5,44,45,46]. The initial template mesh at rest (or T-position) is deformed (registered) to each 3D scan of the dataset [47,48,49] using an optimization algorithm, such as L-BFGS [50]. To parameterize body poses, a kinematic model of the human body is applied. To describe the shape variation in the dataset, principal component analysis (PCA) is applied to the vertices of the deformed meshes [5]. The first ten principal components (shape parameters) are usually used for the mesh regression. Tsoli et al. [6] first registers template meshes to the 3D scans and additionally learns features for body measurement prediction. BUFF [51] addresses the problem of 3D body shape estimation under clothing by allowing the mesh to deviate from the template but regularizes the optimization to satisfy the anthropometric constraints. Similar to non-parametric approaches, 3D parametric approaches are generally more accurate than image-based approaches, but they also require 3D scans as input.

Image-based approaches (2D) can be divided into shape-aware pose estimation methods, which typically regress pose and shape parameters in-the-wild either from 2D keypoints or directly from images [10,11,13,14,15,16,17,52,53,54,55,56,57,58,59,60,61,62,63,64,65,66], and shape estimation methods, which regress shape from silhouettes, usually in fixed pose and minimal clothing [21,22,23,24,25,26,27,28,31,67]. We compare the proposed baseline against the state-of-the-art 3D- and 2D-based approaches for human body measurement estimation and achieve comparable performance to the best methods, while outperforming several deep learning models (see Section 4).

## 3. Method

In this section, we describe the proposed linear regression model and the extraction of body measurements from a template SMPL mesh. We fit separate regression models for males and females. The purpose of the proposed method is to demonstrate that body measurements can be determined using only the information that each person typically knows about themselves, i.e., height and weight, without making any further measurement, which makes it suitable as a baseline.

### 3.1. Linear Regression Model

The method is shown in Figure 1. Each human body sample *i* is defined by 10 shape parameters, $\theta_{Si}$, and 63 pose parameters, $\theta_{Pi}$. The height and the 15 body measurements of the model are extracted from the template mesh. The input to the model consists of height (*h*) and weight (*w*). Weight should be available in the dataset; otherwise, it can be estimated from the calculated mesh volume. The output consists of the 15 other body measurements (A–O). The measurement names are listed in Table 1, and their extraction is described in Section 3.2. Then, the linear model for the *j*-th measurement is described as:(1)$$y_{j}=x^{T}a_{j}+b_{j},\;j\in \left\{1,2,\ldots ,15\right\}$$
where $x^{T}$$\in R^{2}$ is a row vector consisting of the height and weight of the samples (an independent variable), $a_{j}$$\in R^{2}$ is a column vector of the linear coefficients (the slope), $b_{j}\in R$ is an intercept of the regression model, and $y_{j}\in R$ is an output measurement (a dependent variable). These equations can be written more compactly in matrix form as:(2)$$Y_{j}=XA_{j},\;j\in \left\{1,2,\ldots ,15\right\}$$
where $X\in R^{N\times 3}$ are the heights and weights from the *N* samples from the training dataset and $Y_{j}\in R^{N\times 1}$ are their output measurements. Additionally, a column of ones is added to *X* to account for $b_{j}$, which is now included in $A_{j}\in R^{3\times 1}$, representing all the model parameters.

Therefore, the least-squares closed-form solution is:(3)$$A_{j}={\left(X^{T}X\right)}^{-1}X^{T}Y_{j},\;j\in \left\{1,2,\ldots ,15\right\}$$

The linear regression has four assumptions: linearity, homoscedasticity, independence, and normality. The linearity assumes that the relationship between *X* and the mean *Y* is linear. The homoscedasticity assumes that the variance of residual is the same for any value of *X*. Independence assumes that the observations are independent of each other. Finally, the normality assumes that for any fixed value of *X*, *Y* is normally distributed. The latter three assumptions are related to residuals, and we verify those in Section 5.1. For further details on linear regression, we refer readers to the relevant literature [68].

Note that we also experiment with the interaction terms (*h* is height, *w* is weight): $\frac{w}{h^{2}}$ (BMI), wh, $w^{2}$, $h^{2}$, and add them to the input vector $x^{T}$. The model with no interaction terms is called the *baseline*, and the models with added interaction terms are called the *augmented baselines*. The augmented baselines are marked in the remainder of the paper as Baseline (I = N), where *N* is the number of interaction terms. The interaction terms I = 2 correspond to $\frac{w}{h^{2}}$ and wh, and I = 4 corresponds to $\frac{w}{h^{2}}$, wh, $w^{2}$, and $h^{2}$.

In case that the weight measurement is not available in the dataset, we estimate it using the extracted volume. The volume is extracted using a standard algorithm [69]. Then, the body weight (*w*) is estimated based on the human body density, which is approximately ρ=1±0.005 kg/L [31,70], and the extracted volume (*V*). To account for the variation in body density w.r.t. weight, we model the volume as a normal stochastic variable:(4)$$V=V_{extracted}+N\left(\mu_{V}=0,\sigma_{V}=5\right)\;\left[L\right].$$

Note that the standard deviation of 5 L applied to the extracted volume propagates to the standard deviation of 5 kg applied to the estimated weight. Additionally, to account for the variation in self-estimation of height and weight, we model self-estimation using another two stochastic variables, $h=h_{extracted}+N(\mu_{h}\;=\;0,$$\sigma_{h}=1)$ cm, and $w=V\cdot \rho +N(\mu_{w}=0,\sigma_{w}=1.5)$ kg.

### 3.2. Extraction of Body Measurements

We use a total of 18 body measurements, 15 of which are a standard set of measurements used in previous works [21,22,23,24,25,26,27,28,31,67], and 3 of which are used specifically to compare with Virtual Caliper [31] (see Table 1). The measurements are either lengths or circumferences and are calcuated using their corresponding landmarks. The complete list of landmarks with their corresponding SMPL vertex index is shown in Table 2. To extract the lengths, such as the shoulder width and arm length, we calculate the Euclidean distances between the two respective landmarks. To extract the circumferences such as the waist or thigh circumference, we slice the mesh with a horizontal or vertical plane at the specified landmark location and sum up the resulting line segments [8,28].

## 4. Evaluation

We evaluate the linear baseline on BODY-fit [28] and ANSUR [30] datasets. More specifically, the BODY-fit dataset is extended by weight estimations obtained from the extracted mesh volumes. We call the extended dataset BODY-fit+W. We compare the baseline on BODY-fit+W with the following state-of-the-art image-based approaches: SMPLify [10], ExPose [14], and Yan et al. [28]. Unfortunately, most of the other methods have not published their source code or data, so we use their reported results on other datasets instead, such as CAESAR [49] and NOMO3D [8] datasets. In total, six datasets are referenced in this work, as listed in Table 3.

### 4.1. Datasets

The BODY-fit dataset contains 1474 male and 2675 female SMPL meshes. The template meshes are obtained by fitting them to the original 3D scans of people, which are not publicly available. In addition, the BODY-fit+W dataset contains weights that are estimated from corresponding mesh volumes. The distributions of male body measurements on BODY-fit+W, obtained by measuring the template meshes as described in Section 3.2, are shown in Figure 2. The range of the height is between 145 and 196 cm for male and between 135 and 190 cm for female, while weight is between 40 and 130 kg for male, and between 30 and 130 kg for female. Regarding other measurements, they also vary proportionally to their absolute values. For example, ankle circumferences are generally smaller than waist circumferences, etc. The body measurements on BODY-fit+W are similarly diverse as the body measurements from the ANSUR dataset, as shown in Figure 3. The ANSUR dataset is significant, as it represents the realistic human body population, while the BODY-fit+W represents the population of SMPL meshes fitted to 3D scans. Note that not all of the body measurements used in the BODY-fit+W dataset exist in the ANSUR dataset. Table 4 specifies the ANSUR attributes and expressions used to obtain the corresponding measurements from BODY-fit+W.

### 4.2. Quantitative Evaluation

The methods are compared quantitatively against the 15 standard body measurements and the three additional measurements to compare the baseline with the Virtual Caliper [31]. The metrics used for comparison are as follows:Mean absolute error (MAE), $E_{j,MAE}=\frac{1}{N}\sum_{i}^{N}y_{est,j}\left(i\right)-y_{gt,j}\left(i\right)$, where *i* is the sample index, *j* represents the measurement, and *N* is the number of samples;Mean relative error (MRE), $E_{j,MRE}=\frac{1}{N}\sum_{i}^{N}\frac{y_{est,j}\left(i\right)-y_{gt,j}\left(i\right)}{y_{gt,j}\left(i\right)}$, where *i* is the sample index, *j* represents measurement, and *N* is the number of samples;Expert ratio (%<Expert), $\%<Expert_{j}=\frac{\#<Expert_{j}}{N}$, where *j* is the measurement and *N* is the number of samples. This metric shows the ratio of samples that are within the expert errors [30,33,34]. The expert errors are shown in Table 5 and Table 6 (*Expert error* rows).

We compare our linear models against the competing methods in several configurations:Against the methods that use ground-truth features as input, such as ground truth silhouettes [21,22,23,25,26,27]. In this case, we evaluate the baseline using the ground truth volume from the original BODY-fit data, i.e., the volume, height, and weight are modeled as deterministic variables.Against the state-of-the-art methods that use estimated or extracted features, including both 3D-based [6,8,72] and 2D-based [10,11,14,16,21,22,23,25,26,27,28] methods. The volume, height, and weight are modeled as stochastic variables (see Section 3).Against other methods such as the Virtual Caliper [31] that estimates body lengths using a VR headset.More detailed comparison with the representative 2D-based [10] and 3D-based methods [8]. On top of MAE, we also report the mean relative error (MRE) and the percentage of the samples within the expert errors (%<Expert).

**Ground Truth Methods.** We compare the baseline against the methods that use ground truth information as input. Several previous silhouette-based body measurement estimation methods [21,22,23,25,26,27] report their results using the ground truth silhouettes. To evaluate the baseline, we use the BODY-fit dataset and the volume as input. This way, we also exploit the ground truth information instead of estimating the weight. The results are shown in Table 5. Note that several methods [21,22,23] perform within the expert errors. All of these methods, including the baselines, perform comparably and achieve significantly low body measurement errors below 1 cm for all body measurements. However, using ground-truth silhouettes or body volume is unrealistic for real-world anthropometric applications. We provide these analyses for the sake of completeness, following the previous works.

**State-of-the-Art Methods. **Table 6 shows the performance of state-of-the-art body measurement estimation methods, compared to our baselines fitted on the BODY-fit+W and ANSUR datasets. The baseline fitted on the BODY-fit+W dataset models the volume, height, and weight as stochastic variables. For comparison with SMPLify [10,11] and ExPose [14], we scale their meshes to match ground truth height; otherwise, their mesh estimates would be significantly degraded by height estimation errors. Our baseline demonstrates comparable performance with the competing methods, even outperforming several popular deep learning approaches such as HMR [16], SMPLify [10,11], and ExPose [14]. Note that the baseline achieves MAEs within the expert errors for neck circumference (B), shoulder-to-crotch (C), and forearm circumference (I). The baseline evaluated on the ANSUR dataset is less accurate on average, but it still shows competitive performance. A more detailed comparison between BODY-fit+W and ANSUR is given in Section 5.3. The discussion on the performance of deep mesh regression approaches is given in Section 5.4.

**Other Methods.** We also compare to the Virtual Caliper [31] that proposes using a VR headset for body measurement. Compared to our work, the Virtual Caliper evaluates its performance on real subjects and compares it to measurements obtained by their expert, which is more realistic. They also use self-reported weight as input, but they estimate height. For fair comparison, we predict three additional body measurements such as arm span, inseam height, and hip width, as shown in the additional part of Table 1. The results are shown in Table 7. The baseline outperforms the measurements obtained using the Virtual Caliper. The significant advantage of the proposed baseline is that it does not require a VR headset.

**More Detailed Comparison.** To compare the representative 2D- and 3D-based methods to the baseline, we calculate MAE, MRE, and %Expert metrics. The representative 2D-based method is SMPLify [10], even though it does not achieve the best performance among the 2D-based methods (see Table 6). However, the best performing method, by Smith et al. [21], does not provide the source code to evaluate on BODY-fit+W. As shown in Table 8, the 3D-based method by Yan et al. [8], evaluated on the NOMO3D dataset, achieves the best overall performance, with the exception of wrist (G) and ankle circumference (N). The baseline fitted to ANSUR achieves MREs below 5% for all body measurements, which is reasonable and useful for the anthropometric applications. Interestingly, the ratio of samples that are within the expert errors of above 50% are shoulder-to-crotch distance (C) and wrist circumference (G). In general, most ratios are above 25% for most of the body measurements.

## 5. Discussion

The presented baseline demonstrates strong performance on the public datasets. In this section, we analyze the residual hypotheses, *p*-values, and $R^{2}$ scores (Section 5.1), comment on using height and weight for body measurement estimation in more detail (Section 5.2), compare the BODY-fit+W and ANSUR datasets in more detail (Section 5.3), discuss previous image-based mesh regression approaches (Section 5.4), and provide the guidelines for future body measurement estimation methods (Section 5.5).

### 5.1. Residuals, p-Values, and $R^{2}$ Scores

In this section, we verify the three assumptions of linear regression regarding residuals (homoscedasticity, independence, and normality) and check the *p*-values and $R^{2}$ scores for the regression models of each body measurement. Figure 4 shows the residuals for the BODY-fit+W and ANSUR models on train and test splits. The variance of the residuals is generally constant for all the values of both models, which satisfies the homoscedasticity assumption. The values are relatively randomly spread, which satisfies the independence assumption. Finally, as shown in the right sides of the two figures, the means of the residuals are zero and they are normally distributed, which satisfies the normality.

Table 9 and Table 10 show the *p*-values, $R^{2}$ scores, MAEs, and RMSEs for male and female models, on BODY-fit+W and ANSUR, respectively, with two interaction terms (I = 2). We can observe that the vast majority of *p*-values are within the <0.05 threshold. For the simplicity of the analyses, we keep all the input variables that might lead to increased variance in the predictions and hence larger RMSEs [73]. Ideally, the $R^{2}$ scores should be as high as possible. Most of the scores for BODY-fit+W datasets are above or close to 0.8, except for head circumference (A) and shoulder breadth (O). It is reasonable that the head circumference is more difficult to estimate based only on height and weight and their derivative terms. Note that based on $p_{BMI}$ of the shoulder breadth, it would make sense to fit the model without the BMI input term, which may improve the $R^{2}$ score. The ANSUR model has somewhat lower $R^{2}$ scores, particularly for shoulder-to-crotch (C), wrist circumference (G), and ankle circumference (N). Note that the shoulder-to-crotch measure is derived from three manual measurements, as specified in Table 4, which might the part of the reason for the lower score. Intuitively, wrist and ankle circumferences only somewhat correspond to a person’s height and weight, which makes those more difficult to estimate. Finally, even though the linear models could be further improved, the current performance is competitive w.r.t the state-of-the-art 2D- and 3D-based methods, which is the most important observation of these experiments.

### 5.2. Height and Weight for Body Measurement Estimation

It is not completely surprising that height and weight strongly correlate with the body measurements for the statistical model population. The previous works, such as [5,74], analyze the principal components of the SMPL and SCAPE [44] statistical body models, respectively. They add several standard deviations separately to each principal component of the mean shape. The resulting explained variance for the first 10 components is shown in Figure 5, and the variation in shapes are shown in Figure 6. It can be observed that the first two principal components explain most of the variance in body shapes (Figure 5), which in turn define the extreme shape variations visible in Figure 6, particularly in terms of height and weight. Minor variations are visible for the third and fourth components. The remaining components do not significantly influence body shape and measurements. Whether or not these linear relationships hold for the general population, it is currently not easy to verify due to lack of public data. Still, the statistical models are expected to be made from a diverse set of human bodies; therefore, we consider this linear relationship relevant.

### 5.3. Comparing BODY-Fit and ANSUR Models

The two datasets used for the evaluation of the proposed baseline represent the synthetic, statistical population (BODY-fit+W) and the realistic population (ANSUR). The linear models fitted on the BODY-fit+W and ANSUR datasets perform differently for certain body measurements, such as neck circumference (B), shoulder-to-crotch (C), arm length (J), inside-leg length (K), and ankle circumference (N) (see Table 6).

The differences can be explained by the fact that these body measurements are less dispersed in the height–weight space, as shown in Figure 7, even though the volume, height, and weight are modeled as stochastic variables. This might suggest that certain body measurements of the bodies from the synthetic population retain a linear relationship with height and weight, even though Gaussian noise has been added to input. Another reason might be that the human body populations of the two datasets are significantly different w.r.t. to these measurements. ANSUR represents the population of military personnel, while the original population of BODY-fit subjects are probably different.

In order to verify these hypotheses, the actual weight information from BODY-fit subjects is required, at least. Unfortunately, the weight information is not available. Finally, note that we simplify the analysis of ANSUR by not accounting for the diversity in self-estimation of height and weight, because a certain amount of error already exists in the manual measurements, which is difficult to quantify.

### 5.4. On Image-Based Mesh Regression

Most of the previous image-based methods train deep learning models to regress pose and shape parameters that correspond to the template 3D mesh [10,14,16,21,22,23,28], which was made possible with the creation of the statistical models. This approach, while simple and powerful, has the problem of estimating the size of the person in an image. Due to 2D-to-3D scale ambiguity, it is impossible to determine absolute size without additional information [75]. Mesh regression approaches [10,14,16] typically ignore this problem by training convolutional networks on images and hoping that the network will fit proper height by exploiting the image context. As a result, they often estimate an incorrect absolute height when not explicitly trained for scale estimation. To allow a fair comparison with these methods, we scale their output meshes to match the ground truth height by multiplying with the ratio between the mesh height and the ground truth height. Nevertheless, errors are propagated to the body measurements, which is probably because the body measurements are distributed differently for different body heights. On the other hand, silhouette-based methods [21,22,23,28] typically render human meshes at a fixed distance from the camera and then train their model to estimate the height. The drawback of this approach is that the model only estimates the height correctly for that particular camera–distance configuration.

### 5.5. Limitations, Assumptions, and Future Guidelines

**Limitations.** The major limitation of body measurement estimation in general is a lack of large, public, and realistic benchmarks. The public benchmark used in this work, based on BODY-fit [28], contains 3D template meshes fitted to 3D scans. Even though the fitted meshes represent the scanning dataset, the SMPL fits are still an approximation of the original scans. Moreover, the 3D scanning process is also not perfect [9], so the scans themselves do not necessarily represent the original physical human bodies.

**General Assumptions.** Therefore, the first assumption is that the manual body measurements are comparable to the body measurements obtained from the 3D scan, i.e., from the SMPL template mesh. In addition, all ANSUR body measurements do not necessarily physically correspond to the measurements from the SMPL model, such as shoulder-to-crotch or waist circumference, as specified in Section 4.1, but we assume they are highly linearly correlated and thus comparable. The height is measured as the difference between the top head point and the heel point on the *y* (height) axis. However, most of the datasets’ subjects are expected to take approximately the A-pose, which is not fully erect. This might result in height being incorrectly estimated in some cases. We assume that the posture is not significantly affected by breathing, that the subjects wear minimal or tight clothes, and that the body measurements such as height and head circumference are not affected by hair artifacts. The problem with hair artifacts is particularly important for the female subjects. Finally, we model all stochastic variables by adding normal distributions that are uncorrelated with other variables, which is not realistic for all groups inside the population. For example, underweight people tend to overestimate their weight and vice versa [76].

**Future Guidelines.** To be able to evaluate our model, as well as other body measurement estimation models, even more reliably, future work should focus on creating more realistic, diverse, and public benchmarks. To avoid estimating body weight from volume, the benchmark and the training dataset should contain measured weights. For the evaluation of future methods, we propose the 15 body measurements as a basic set of measurements for mesh comparison. The body measurements are intuitive and simple to obtain from template mesh, such as SMPL. The evaluation of the 15 measurements can be used in addition to the previously used per-vertex error (PVE) metric, which is used to compare between the mesh regression methods [12,61,77]. For more standardized body measurement comparison, we specify the exact vertices used to obtain each body measurement from the SMPL mesh.

**Measurement Standards.** When creating anthropometric datasets, special attention must be paid to body measurement standards [33] and body postures [34]. Even though manual expert measurements are considered to be the golden standard, they will never be perfect, because the human body is not rigid [78]. There are many proposed methods for body measurement error analysis [79,80,81,82], but the conclusion is that no statistical procedure is optimal for manual anthropometry in general. [83]. The most common, simple to calculate, and easily interpretable approach, proposed by Gordon et al. [30], includes the allowable or “expert” errors [30], which we mentioned earlier and compare against in Table 5 and Table 6. The authors took into account many factors that potentially affect measurement accuracy, such as posture, time of the day (morning, evening), measurement technique and instrument, etc. The allowable errors are the inter-observer values obtained by the expert measurers based on the mean of absolute differences. Those values have been incorporated into the international standards [33,34].

**Synthetic Data.** Finally, the advantage of using synthetic data and 3D template meshes is that the body measurements can be extracted in a standardized way, which guarantees a standardized measurement. Another advantage is that synthetic data avoid privacy issues and approvals. Therefore, the promising future direction is the creation of large, more realistic, and more diverse synthetic anthropometric benchmarks. The statistical models such as SMPL are currently the best approximation of the overall population and are likely to be used as a tool for generating the body measurement benchmarks in the coming years.

### 5.6. Implementation Details

We implement the model and the experiments in Python 3.8. For linear regression and other models in Section 5, we use the scikit-learn package [84]. For creating mesh objects, calculating volume, slicing with planes, and visualization, we use the trimesh package [85]. For processing 3D meshes as SMPL template meshes and calculating body measurements, we use part of the public SMPL-X source code [11]. For other efficient computations, we use NumPy [86] and SciPy [87]. The seed for generating random numbers is set to (seed=2021). All experiments are done on a single desktop computer, with an Intel Core i7-9700 CPU (3GHz, 8-core), 16GB RAM, and NVidia GeForce RTX 2080 Super, under Ubuntu 20.04 LTS. The source code, demos, and instructions are attached as Appendix A.

## 6. Conclusions

The presented regression method is a simple but significantly accurate tool for the convenient and automatic estimation of body measurements, without the need to remove clothes or capture images. Our work demonstrates that the self-reported height and body weight predict body dimensions as good as the state-of-the-art deep learning methods, or even better. The linear regression based on self-reported height and weight should be used as the baseline for future methods; i.e., any body measurement estimation method should never perform worse. The results reported in this paper mostly serve to show the approximate relative performances between the baseline and the competing methods on the currently available public benchmarks, but for the future reference and datasets, we recommend fitting the model to more diverse and realistic data, if applicable. The baseline is perfectly suitable for but not limited to the applications such as virtual try-on, VR, or ergonomics. The next step is to create more realistic, statistical model-based, public benchmarks to further evaluate the relationship between height and weight as well as other anthropometric measurements.

## Figures and Tables

**Figure 1 sensors-22-01885-f001:**
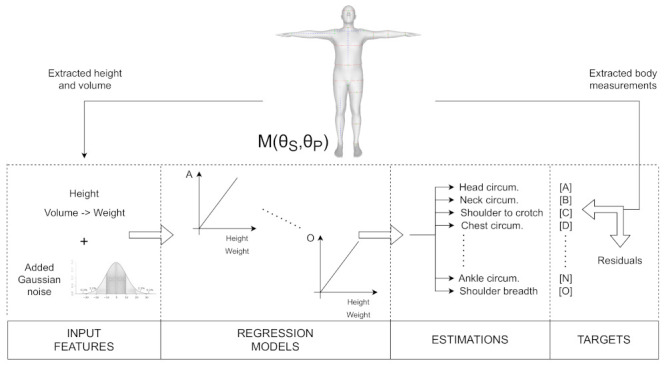
An overview of the linear regression model for a single sample from the statistical model. The sample mesh is defined using the shape and pose parameters, $M{\left(\theta_{S},\theta_{P}\right)}_{i}$. Both input and output are extracted from the sample template mesh, $M_{i}$. The input consists of height and weight. Weight can be either available in the dataset or estimated using the calculated mesh volume. The output consists of the 15 body measurements (A–O), which are listed in Table 1. To account for the errors in self-reporting height and weight, we model height and weight as stochastic variables by adding Gaussian noise to the input. A linear regression model is fitted to each of the 15 body measurements.

**Figure 2 sensors-22-01885-f002:**
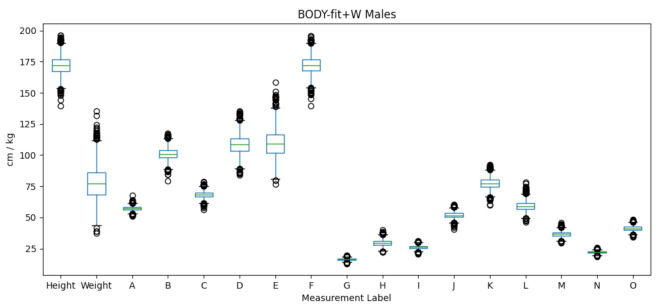
The distribution of body measurements for male subjects in the BODY-fit+W dataset.

**Figure 3 sensors-22-01885-f003:**
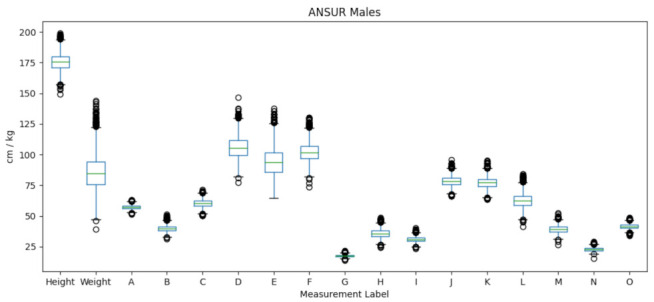
The distribution of body measurements for male subjects in the ANSUR dataset.

**Figure 4 sensors-22-01885-f004:**
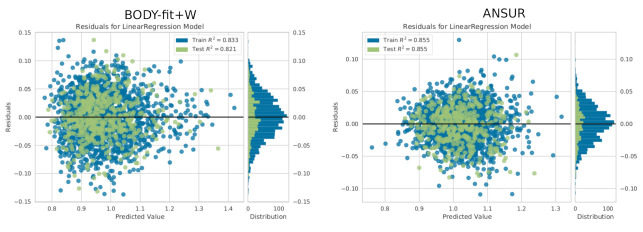
An analysis of the residuals for hip circumference (F), for BODY-fit+W and ANSUR, on train and test splits.

**Figure 5 sensors-22-01885-f005:**
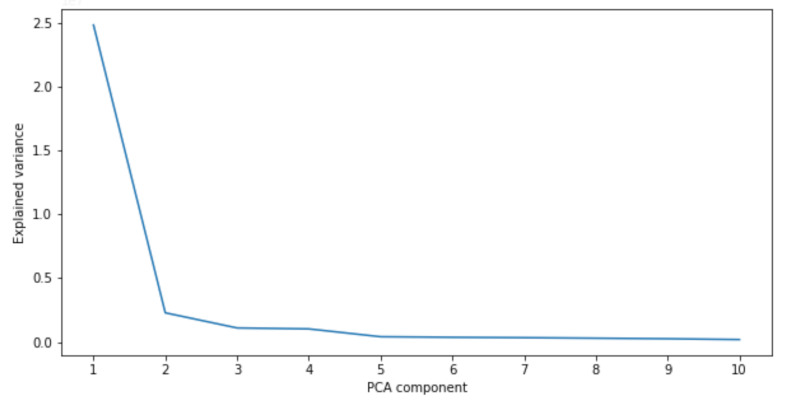
Explained variance for the first 10 principal components of the dataset. The graph is generated on CAESAR-fits data [74] by applying PCA to the given vertices. As expected, the first two components, which most significantly correlate to height and weight, explain most of the variance in the data.

**Figure 6 sensors-22-01885-f006:**
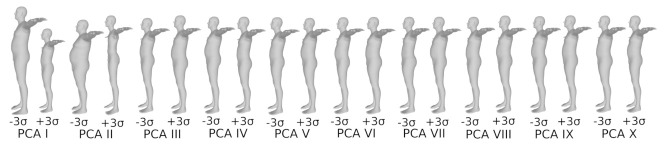
Explained variance for the first 10 principal components of the dataset. As expected, the first two components, which highly correlate to height and weight, explain most of the variance in the data. The image is inspired by [74], originally made using SCAPE model [44]. We have drawn the above image using SMPL model [5].

**Figure 7 sensors-22-01885-f007:**
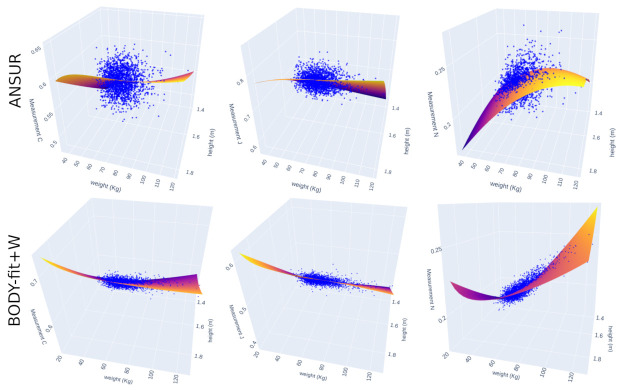
Out of all measurements, the above images show the extreme three cases of body measurements with different dispersions on the ANSUR (**first row**) and BODY-fit+W (**second row**) datasets: shoulder-to-crotch (C), arm length (J), and ankle circumference (N). The fitted planes correspond to the linear model with two interaction terms (I = 2).

**Table 1 sensors-22-01885-t001:** The list of 15 (+3) body measurements. The 15 measurements (A–O) are used to compare to the state-of-the-art. The arm length (J) and the three additional measurements are specifically used to compare with [31].

Measurement Set		Measurement	Landmark Index
Standard	A	Head circumference	14
	B	Neck circumference	10
	C	Shoulder to crotch	1, 10
	D	Chest circumference	4
	E	Waist circumference	13
	F	Hip circumference	19
	G	Wrist circumference	9
	H	Bicep circumference	20
	I	Forearm circumference	15
	J	Arm length	2, 9
	K	Inside leg length	11, 12
	L	Thigh circumference	16
	M	Calf circumference	17
	N	Ankle circumference	18
	O	Shoulder breadth	2, 3
Additional [31]	-	Arm span	7, 8
	-	Inseam height	2, 19
	-	Hip width	5, 6

**Table 2 sensors-22-01885-t002:** The list of 20 landmarks and their corresponding SMPL vertex indices.

Landmark Index	Landmark Name	Vertex Index	Landmark Index	Landmark	Vertex Index
1	Inseam point	3149	11	Low left hip	3134
2	Left shoulder	3011	12	Left ankle	3334
3	Right shoulder	6470	13	Lower belly point	1769
4	Left chest	1423	14	Forehead point	335
5	Left hip	1229	15	Right forearm point	5084
6	Right hip	4949	16	Right thigh point	4971
7	Left mid finger	2445	17	Right calf point	4589
8	Right mid finger	5906	18	Right ankle point	6723
9	Left wrist	2241	19	Mid hip point	3145
10	Shoulder top	3068	20	Right bicep point	6281

**Table 3 sensors-22-01885-t003:** The list of datasets referenced in this work. Note that the baseline is evaluated on the first three datasets, above the dashed line.

Dataset	Samples	Data Type	Availability	Approach	Reported by
BODY-fit	4149	SMPL mesh	Public	2D-based	[28], Our
BODY-fit+W	4149	SMPL mesh	Public	2D-based	[10,14,28], Our
ANSUR	6068	Tabular	Public	Regression	ISO [30], Our
CAESAR	3800	Point cloud	Proprietary	3D-based	[21,23,25,26,27]
					[6,16,22,71,72]
NOMO3D	375	Point cloud	Public	3D-based	[8]
Virtual Caliper	20	Point cloud	Private	Regression	[31]

**Table 4 sensors-22-01885-t004:** The specification of ANSUR attributes and expressions corresponding to the body measurements extracted from the SMPL meshes.

	SMPL Mesh (BODY-fit+W)	ANSUR Attribute/Expression
A	Head circumference	headcircumference
B	Neck circumference	neckcircumference
C	Shoulder to crotch	sittingheight-(stature-acromialheight)
D	Chest circumference	chestcircumference
E	Waist circumference	waistcircumference
F	Hip circumference	buttockcircumference
G	Wrist circumference	wristcircumference
H	Bicep circumference	bicepcircumferenceflexed
I	Forearm circumference	forearmcircumferenceflexed
J	Arm length	acromialheight-wristheight
K	Inside leg length	crotchheight-lateralmalleolusheight
L	Thigh circumference	thighcircumference
M	Calf circumference	calfcircumference
N	Ankle circumference	anklecircumference
O	Shoulder breadth	biacromialbreadth

**Table 5 sensors-22-01885-t005:** Quantitative comparison to image-based body measurement approaches that use *ground truth* silhouettes (MAEs, in mm). In this case, we also show the performance of the linear baseline using the volume, height, and weight as deterministic variables (unrealistic). Note that we demonstrate only the performance of the BODY-fit model, as ANSUR does not contain volume measurements. The baseline (I = 4) additionally uses four interaction terms, as described in Section 3. Methods marked with † are evaluated on different, non-public data, and the results are reported in [21]. The best results are shown in bold.

Measurement	Dataset	A	B	C	D	E	F	G	H	I	J	K	L	M	N	O	Mean
†Xi ’07 [27]	CAESAR	50.0	59.0	119	36.0	55.0	23.0	56.0	146	182	109	19.0	35.0	33.0	61.0	24.0	67.1
†Chen ’10 [26]	CAESAR	23.0	27.0	52.0	18.0	37.0	15.0	24.0	59.0	76.0	53.0	9.0	19.0	16.0	28.0	12.0	31.2
†Boisvert ’13 [25]	CAESAR	10.0	11.0	4.0	10.0	22.0	11.0	9.0	17.0	16.0	15.0	6.0	9.0	6.0	14.0	6.0	11.1
Expert error [30]	ANSUR	5.0	6.0	15.0	12.0	12.0	-	-	-	6.0	-	4.0	-	-	-	8.0	8.5
†Dibra ’17 [23]	CAESAR	3.2	1.9	4.2	5.6	7.1	6.9	1.6	2.6	2.2	2.3	4.3	5.1	2.7	**1.4**	2.1	3.6
†Dibra ’16 [22]	CAESAR	**2.0**	2.0	3.0	**2.0**	7.0	4.0	2.0	2.0	**1.0**	3.0	9.0	6.0	3.0	2.0	2.0	3.3
†Smith ’19 [21]	CAESAR	5.1	3.0	**1.5**	4.7	**4.8**	**3.0**	2.5	2.7	1.9	**1.7**	**1.5**	**2.4**	**2.3**	2.1	**1.9**	**2.7**
Baseline (I = 4)	BODY-fit	7.9	**1.2**	5.6	10.1	9.2	3.6	**0.6**	**1.4**	1.3	5.3	8.0	8.4	2.6	**1.4**	6.6	4.9

**Table 6 sensors-22-01885-t006:** Quantitative comparison to image-based body measurement and shape estimation approaches (MAEs in mm). In this case, we show the performance of the linear baseline using self-reported height and weight as stochastic variables (more realistic). The BODY-fit+W and ANSUR datasets are used to present the baseline performance. The baseline used I = 2 interaction terms, as described in Section 3. The best results are shown in bold.

Method	Dataset	A	B	C	D	E	F	G	H	I	J	K	L	M	N	O	Mean
HMR [16]	CAESAR	16.7	35.7	33.8	92.8	118	68.7	12.2	29.3	20.6	29.9	44.3	38.5	25.8	14.0	26.5	39.8
ExPose [14]	BODY-fit	17.4	13.1	31.4	96.0	116.7	54.8	7.7	33.3	15.3	12.3	29.5	37.3	18.2	8.9	23.0	34.3
SMPLify [10]	BODY-fit	15.3	7.7	8.7	57.5	74.7	39.7	5.1	21.0	9.5	5.7	11.4	27.2	12.3	6.5	10.4	21.6
Hasler ’09 [71]	CAESAR	7.5	17.0	7.5	13.0	19.0	16.2	-	-	-	10.4	-	-	-	6.6	-	12.2
Anthroscan [72]	CAESAR	7.4	21.1	7.5	12.4	17.0	7.5	-	-	-	11.7	-	-	-	7.6	-	11.5
Tsoli ’14 [6]	CAESAR	5.9	15.8	5.5	12.7	18.6	12.4	-	-	-	10.1	-	-	-	6.2	-	10.9
Yan ’20 [28]	BODY-fit	12.0	13.6	8.9	22.2	16.9	14.2	4.8	10.0	8.0	6.8	7.5	13.8	9.1	5.9	8.2	10.8
Dibra ’16 [22]	CAESAR	9.3	10.0	6.6	22.8	24.0	20.0	9.9	12.0	7.9	6.4	8.9	15.5	13.2	7.6	6.0	10.7
Expert Error [30]	ANSUR	**5.0**	6.0	15.0	**12.0**	**12.0**	-	-	-	6.0	-	4.0	-	-	-	8.0	8.5
Yan ’20 [8]	NOMO3D	-	**3.7**	-	13.2	12.4	**8.9**	4.5	**5.5**	**3.0**	13.2	-	**7.9**	**3.0**	10.6	12.4	8.2
Smith ’19 [21]	CAESAR	6.7	8.0	**5.1**	12.5	15.8	9.3	9.3	8.1	5.7	**5.1**	**6.8**	8.8	7.2	5.0	**4.5**	**7.9**
Baseline (I = 2)	BODY-fit+W	9.1	4.2	6.6	30.3	39.5	28.0	**2.7**	10.0	4.9	5.7	9.5	16.0	7.3	**3.3**	9.0	12.4
Baseline (I = 2)	ANSUR	11.9	10.7	17.4	29.1	37.9	21.6	4.4	13.2	9.3	17.6	19.6	17.0	12.8	8.7	11.4	16.2

**Table 7 sensors-22-01885-t007:** Comparison to the Virtual Caliper [31] (MAEs in mm) w.r.t. four body measurements—arm length (J) and the three additional measurements (arm span, inseam height, and hip width). We present the baseline evaluated on the same data as in Table 6 (BODY-fit+W). Better results are shown in bold.

Measurement	Dataset	Arm Span	Arm Length	Inseam Height	Hip Width	Mean
Virtual Caliper [31]	Virtual Caliper	17.2	7.6	24.6	**6.5**	14.0
Baseline (I = 2)	BODY-fit+W	**13.1**	**5.7**	**8.8**	6.7	**8.6**

**Table 8 sensors-22-01885-t008:** Detailed comparison between 2D-based methods (SMPLify [10]), 3D-based methods (Yan et al. [8]), and the two linear baselines (with I = 2 interaction terms), one fitted to the BODY-fit+W dataset, and one fitted to the ANSUR dataset [30]. Note that to fairly compare with Yan et al., the expert error values are extended according to [8]. The best results in each row for MAEs and %<Experts are shown in bold.

	2D-Based			3D-Based			Baseline (I = 2)			Baseline (I = 2)		
	SMPLify [10] (BODY-fit+W)			Yan et al. [8] (NOMO3D)			BODY-fit+W			ANSUR		
	**MAE [mm]** **↓**	**MRE (%)** **↓**	**%<Expert ↑**	**MAE**	**MRE**	**%<Expert**	**MAE**	**MRE**	**%<Expert**	**MAE**	**MRE**	**%<Expert**
A	15.3	2.3	25.1	-	-	-	**9.1**	1.5	**43.1**	11.9	2.1	27.9
B	7.7	4.4	50.7	**3.7**	-	87.6	4.2	1.1	74.9	10.7	2.9	34.9
C	8.7	1.4	85.2	-	-	-	**6.6**	0.9	**94.2**	17.4	3.0	50.1
D	57.5	5.5	13.6	**13.2**	-	**67.6**	30.3	1.4	23.8	29.1	2.9	27.3
E	74.7	7.0	9.3	**12.4**	-	**58.7**	39.5	1.6	19.8	37.9	4.2	20.6
F	39.7	5.9	12.3	**8.9**	-	**72.4**	28.0	1.1	23.4	21.6	2.1	35.8
G	5.1	3.3	59.5	4.5	-	66.5	**2.7**	0.7	**87.0**	4.4	2.7	63.2
H	21.0	7.5	17.2	**5.5**	-	**65.8**	10.0	1.4	33.8	13.2	4.0	28.5
I	9.5	3.8	40.0	**3.0**	-	**74.2**	4.9	0.9	63.9	9.3	3.2	40.1
J	**5.7**	1.6	-	13.2	-	-	**5.7**	1.2	-	17.6	2.3	-
K	11.4	1.7	21.0	-	-	-	**9.5**	1.4	**26.8**	19.6	2.6	13.9
L	27.2	4.5	14.5	**7.9**	-	**47.5**	16.0	1.7	23.4	17.0	2.7	25.1
M	12.3	3.4	27.5	**3.0**	-	**82.5**	7.3	1.0	40.7	12.8	3.4	25.5
N	6.5	2.9	41.4	10.6	-	26.7	**3.3**	0.8	**60.5**	8.7	8.7	28.0
O	10.4	3.2	49.2	12.4	-	-	**9.0**	1.8	**56.2**	11.4	2.9	43.1

**Table 9 sensors-22-01885-t009:** The linear regression statistics for the BODY-fit+W dataset, for males and females (I = 2). The *p*-values correspond to the intercept (*b*), height (*h*), weight (*w*), BMI ($\frac{w}{h^{2}}$), and wh, respectively. In addition, we report adjusted $R^{2}$ scores, MAEs (mm), and RMSEs (mm).

	MALE								FEMALE							
	$p_{b}$	$p_{h}$	$p_{w}$	$p_{\mathrm{BMI}}$	$p_{\mathrm{wh}}$	**Adj.** $R^{2}$	**MAE**	**RMSE**	$p_{b}$	$p_{h}$	$p_{w}$	$p_{\mathrm{BMI}}$	$p_{\mathrm{wh}}$	**Adj.** $R^{2}$	**MAE**	**RMSE**
A	0.004	0.000	0.000	0.000	0.000	0.508	8.90	12.36	0.000	0.000	0.000	0.000	0.000	0.442	8.31	12.67
B	0.000	0.255	0.000	0.033	0.000	0.796	4.00	5.04	0.000	0.000	0.000	0.000	0.000	0.795	4.12	5.22
C	0.351	0.000	0.000	0.000	0.000	0.892	6.95	8.77	0.001	0.000	0.000	0.000	0.000	0.903	6.16	7.73
D	0.402	0.000	0.000	0.067	0.000	0.805	26.68	33.81	0.053	0.000	0.000	0.000	0.000	0.808	31.81	40.29
E	0.094	0.000	0.000	0.000	0.000	0.811	38.55	49.18	0.000	0.000	0.000	0.000	0.000	0.819	38.27	48.95
F	0.904	0.000	0.000	0.001	0.000	0.829	22.25	28.50	0.000	0.000	0.000	0.000	0.000	0.833	30.77	39.45
G	0.264	0.000	0.000	0.001	0.000	0.845	2.74	3.46	0.52	0.000	0.000	0.000	0.000	0.85	2.44	3.14
H	0.718	0.005	0.000	0.055	0.000	0.811	8.46	10.76	0.000	0.000	0.000	0.000	0.000	0.825	10.47	13.40
I	0.019	0.000	0.000	0.354	0.000	0.841	4.44	5.71	0.000	0.000	0.000	0.000	0.000	0.848	4.99	6.44
J	0.489	0.000	0.421	0.281	0.359	0.930	5.81	7.81	0.000	0.000	0.055	0.648	0.055	0.923	6.07	8.19
K	0.600	0.000	0.000	0.128	0.001	0.903	10.10	13.26	0.000	0.000	0.007	0.009	0.111	0.920	8.91	11.51
L	0.580	0.000	0.000	0.021	0.002	0.742	14.01	18.70	0.846	0.000	0.000	0.000	0.000	0.790	8.91	21.57
M	0.011	0.000	0.000	0.113	0.000	0.810	7.10	9.29	0.012	0.000	0.000	0.000	0.000	0.835	6.69	8.70
N	0.257	0.000	0.000	0.000	0.000	0.856	2.76	3.49	0.925	0.000	0.000	0.000	0.000	0.848	3.17	4.11
O	0.000	0.034	0.000	0.980	0.015	0.679	8.54	10.78	0.000	0.000	0.000	0.000	0.000	0.689	8.45	10.81

**Table 10 sensors-22-01885-t010:** The linear regression statistics for the ANSUR dataset, for males and females (I = 2). The *p*-values correspond to the intercept (*b*), height (*h*), weight (*w*), BMI ($\frac{w}{h^{2}}$), and wh, respectively. In addition, we report adjusted $R^{2}$ scores, MAEs (mm), and RMSEs (mm).

	MALE								FEMALE							
	$p_{b}$	$p_{h}$	$p_{w}$	$p_{\mathrm{BMI}}$	$p_{\mathrm{wh}}$	**Adj.** $R^{2}$	**MAE**	**RMSE**	$p_{b}$	$p_{h}$	$p_{w}$	$p_{\mathrm{BMI}}$	$p_{\mathrm{wh}}$	**Adj.** $R^{2}$	**MAE**	**RMSE**
A	0.000	0.011	0.116	0.014	0.349	0.266	10.50	13.25	0.000	0.370	0.654	0.002	0.642	0.152	13.19	17.02
B	0.000	0.863	0.000	0.030	0.001	0.671	12.01	15.09	0.002	0.374	0.000	0.902	0.028	0.606	9.45	12.14
C	0.262	0.000	0.185	0.312	0.402	0.484	16.90	21.70	0.635	0.000	0.234	0.779	0.263	0.405	17.90	21.96
D	0.000	0.137	0.000	0.000	0.000	0.866	24.46	31.34	0.049	0.632	0.000	0.400	0.001	0.740	33.96	43.56
E	0.000	0.123	0.000	0.080	0.000	0.848	36.50	45.31	0.344	0.804	0.000	0.440	0.000	0.777	39.37	49.78
F	0.000	0.574	0.000	0.000	0.001	0.887	20.79	26.68	0.000	0.892	0.000	0.000	0.026	0.855	22.34	28.38
G	0.140	0.000	0.000	0.548	0.000	0.547	4.90	6.15	0.001	0.020	0.300	0.015	0.826	0.543	3.89	4.90
H	0.000	0.000	0.122	0.000	0.370	0.712	15.26	19.72	0.019	0.625	0.000	0.249	0.001	0.800	11.09	14.11
I	0.000	0.435	0.014	0.000	0.696	0.662	10.65	13.44	0.000	0.779	0.004	0.005	0.182	0.678	8.03	10.15
J	0.592	0.000	0.054	0.209	0.091	0.649	15.59	19.39	0.323	0.000	0.875	0.862	0.830	0.646	14.29	18.05
K	0.017	0.000	0.265	0.440	0.478	0.714	19.60	25.00	0.121	0.000	0.605	0.341	0.644	0.692	19.63	24.11
L	0.000	0.000	0.076	0.000	0.026	0.859	17.12	22.15	0.000	0.000	0.109	0.000	0.402	0.840	16.71	21.59
M	0.000	0.272	0.000	0.000	0.228	0.711	12.28	15.62	0.000	0.063	0.587	0.000	0.381	0.648	13.35	16.62
N	0.000	0.029	0.004	0.000	0.154	0.533	8.07	10.28	0.000	0.494	0.470	0.000	0.210	0.385	9.31	11.61
O	0.072	0.000	0.000	0.808	0.004	0.420	11.26	14.09	0.054	0.000	0.845	0.016	0.883	0.373	11.49	14.50

## Data Availability

The benchmark data is available as a part of Appendix A.

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
