# Peer review of "Linear Regression vs. Deep Learning: A Simple Yet Effective Baseline for Human Body Measurement"

_sensors, 2022, doi:10.3390/s22051885_

Round 1

Reviewer 1 Report

In the revised version, the authors have added more details and descriptions on the experiments and results.

However, the authors should pay more attention on the format of the tables, some of which are too big. Besides, the subtitles of the figures are not that clear when zoom-in...

So the details should be checked before published.

Author Response

We thank the reviewer for the comments. We update Figure 6 labels. Please note that the tables that are large are still within the template's margins. If this is not in accordance with the MDPI template, please help us by suggesting what to do with the tables. We could decrease the font, but this might sacrifice readability.

Reviewer 2 Report

As someone with a long career in anthropometry, I struggled to understand the point of this paper. Ultimately, what is presented are a set of simplistic linear regression models predicting selected body dimensions from stature and body weight. People in the anthropometry community have been imputing dimensions from overall body characteristics for decades using detailed datasets of manual and (more recently) scan-extracted dimensions, so this idea is far from novel.

But on further reflection I decided that this must be an “emperor has no clothes” paper that is claiming that widely used methods for estimating body dimensions from 2D and 3D data have surprisingly bad performance compared with a simple approach based on manually measured (and potentially self-reported) anthropometry.

I have a variety of comments and suggestions that will make this point more clearly (if indeed that is the point) and also to better elucidate how the current results could be built upon for anthropometric applications.

  1. It is critical to point out that the dimensions used in this analysis are not strictly comparable. Specifically, none of the dimensions are measured in a manner that is “ANSUR equivalent”, i.e., can reasonably be compared with values measured using standard manual anthropometric techniques. To take a simple example, head circumference in 3D scan data is contaminated by hair artifacts; a circumference taken on a scan, even one obtained using a wig cap or some other method to compress the hair, will not be equivalent to manual measurements with a tape.
  2. When the dimensions are not comparable, the error analyses are not either. For example, a regression predicting scan-extracted chest circumferences from overall body dimensions can’t be expected to produce results equivalent to a dimension measured with a tape. The developers of products like Anthroscan have worked diligently to improve the comparability of scan-extracted and manual measurements, but, ultimately, they are simply different measurements (the postures are always different, for example). This point should be made explicitly in the paper.
  3. Please list the specific dimensions you are using for each study. For example, ANSUR88 does not have “hip circumference”. Do you mean buttock circ?
  4. I have not been able to duplicate the mean absolute errors using regressions (with your four “interaction” terms) in the ANSUR88 dataset. The MAE for these regressions (male only) are (mm):

HEAD_CIRC 10.5

NECK_CIRC.BASE 11.

CHEST_CIRC 24.1

WAIST_CIRC.OMPHALION 33.1

            (Note that I had to guess which dimensions you had in mind since you do not specify which ANSUR or CAESAR dimension you used.) Comparing to Table 4, which I assume lists MAE (the table caption does not specify the metric), I see very large discrepancies. More generally, based on my familiarity with anthropometric data, including a large amount of experience with imputing dimensions from other dimensions, I find your error values to be implausibly small, if in fact these are MAE. Please help me to understand how you achieved these results. Showing the performance of the regressions within ANSUR and/or CAESAR standard dimensions alone would go a long way to helping to clarify what you’re doing. [Note your reference to Gordon et al. has the authors and title of ANSUR II (2012) but a date of the publication of ANSUR88 (i.e., 1989). Which dataset did you use? It won’t affect the results much, but you should be clear, particularly because the definitions of a few measurements changed.]

  1. Your presentation of the regressions is inadequate, particularly since this is the focus of the paper. Please present the full regression models (all coefficients for I=4 and I=2) along with p values, adjusted R^2 values, MAE (since that’s your preferred error metric) and root-mean-square errors for regressions with male & female data.
  2. Please present additional discussion, with references, of the validity and likely errors in self-reported stature and weight. Modeling with a random component (uncorrelated?) would be expected to inflate your errors, but as noted above they are implausibly small as-is.
  3. Note that scan-extracted stature is not equivalent to manually measured stature. Values tend to be inflated by hair artifacts and reduced by a scan posture that is different from the maximally erect posture used to record stature. Please include this fact and the likely effects on your analysis in your presentation of the actual dimensions you used for each study (particularly when you normalized silhouette- and scan-based data to “ground truth” stature) and differences between scan-extracted and manual measurements.
  4. Please incorporate a more meaningful discussion (at least a paragraph) of allowable error. Gordon et al. (2012) has an excellent discussion, with references and procedures, in Section 7. Since you’re comparing results to these values, it’s critical that you help the reader to understand what these “expert errors” actually mean.

Author Response

We want to thank the reviewer for particularly detailed comments and suggestions on how to further improve our work. We took all the suggestions into account and updated our manuscript, as well as responded to all the concerns in the remainder of this document. 

R1) As someone with a long career in anthropometry, I struggled to understand the point of this paper. Ultimately, what is presented are a set of simplistic linear regression models predicting selected body dimensions from stature and body weight. People in the anthropometry community have been imputing dimensions from overall body characteristics for decades using detailed datasets of manual and (more recently) scan-extracted dimensions, so this idea is far from novel. 

A1) We agree with the reviewer. The published literature is abundant with papers which all have in common the very same underlying principle for anthropometry estimation. In general, they either use a standard approach or deep learning approach where in either case, as the reviewer noticed, the input is “dimensions from overall body characteristics” and from that perspective we only follow the previous works. Our main motivation, on the other hand, is to additionally show that a simple linear regression model, based only on height and weight, demonstrates strong performance w.r.t. to much more powerful state-of-the-art 2D- and 3D-based methods. 

R2) But on further reflection I decided that this must be an “emperor has no clothes” paper that is claiming that widely used methods for estimating body dimensions from 2D and 3D data have surprisingly bad performance compared with a simple approach based on manually measured (and potentially self-reported) anthropometry. 

A2) The reviewer is right. One of the main purposes of the paper is to show that the simple approach is comparable to the widely used ones, but much more complex methods for body measurement estimation. We agree that the presented approach has several important assumptions, many of which the reviewer has pointed out. Please note that some of these assumptions, such as the one that the body measurements acquired manually or using 3D scanning followed by statistical model fitting are equivalent, are required in order to compare the results obtained using different methods and on different data. Even more importantly, all the competing 2D and 3D-based methods exploit these exact same assumptions, which we particularly discuss in the Section 5.3 of the original version.  

Please allow us to give you an explicit example. The work by Smith et al. [22], the state-of-the-art method in 2D-based body measurement, uses both known height and known weight, as well as the whole deep learning model to fit 3D meshes to input images. Our linear regression model achieves similar performance using only height and weight. 

We are also aware that there are many praiseworthy previous works that present and analyze their linear regression models under much greater rigor, analyzing the groups of people inside the dataset population, and considering the statistics of the particular groups. For example, underweight people tend to overestimate their weight and vice versa. However, we still argue that our model is a simple and straightforward tool to rather accurately assess body measurements based on minimal input information. Due to its simplicity, the model will not be applicable to all groups.  

Finally, it is well known that current deep learning models very often lack the desired formality, interpretations, and explanations. One could also see our model as a “naked” deep learning-like approach, using the most fundamental information. By using only the crucial information, we provide some insight(s) on the performance of more complex, competing approaches, and these insights can be used as a starting point, i.e., the baseline. 

R3) It is critical to point out that the dimensions used in this analysis are not strictly comparable. Specifically, none of the dimensions are measured in a manner that is “ANSUR equivalent”, i.e., can reasonably be compared with values measured using standard manual anthropometric techniques. To take a simple example, head circumference in 3D scan data is contaminated by hair artifacts; a circumference taken on a scan, even one obtained using a wig cap or some other method to compress the hair, will not be equivalent to manual measurements with a tape. 

A3) “Contamination” by hair affect is certainly present but not to the extent that data could not be used. Many people have a rather short hair or no hair at all. In particular, please note that ANSUR data set is based on the military personnel which in most cases have a rather short hair. In such cases, measuring directly or using a wig cap to compress the hair should not be an argument to completely discard data. Finally, most other ANSUR measurements are not affected by the hair. Nevertheless, following the reviewers remark we mention the hair issue in our paper (Section 5.4), as well as several other specific assumptions related to 3D-scan-to-manual-measurement data, such as tight- or no-clothing assumption, etc. 

R4) When the dimensions are not comparable, the error analyses are not either. For example, a regression predicting scan-extracted chest circumferences from overall body dimensions can’t be expected to produce results equivalent to a dimension measured with a tape. The developers of products like Anthroscan have worked diligently to improve the comparability of scan-extracted and manual measurements, but, ultimately, they are simply different measurements (the postures are always different, for example). This point should be made explicitly in the paper. 

A4) The reviewer is right. There are always differences between manual and 3D scan (automatic) measurements. In fact, there are differences between manual measurements themselves when performed by different people or even the same person on different occasions (the so-called “expert” or “allowable” error). Therefore, as the reviewer pointed out, the (absolute) errors between such measurements are also not comparable, but they are comparable up to some tolerance, hence the allowable errors. It is true that we do not diligently work to improve the comparability between the manual and 3D scan measurements, but we do specify the exact indices used to obtain the body measurements from template mesh, which should improve reproducibility of our approach and, hopefully, future works. Regarding the postures, all the samples are expected to be fully erect, which simplifies the body measurement process. As requested by the reviewer, we explicitly point out the comparability assumption (Section 5.5).  

R5) Please list the specific dimensions you are using for each study. For example, ANSUR88 does not have “hip circumference”. Do you mean buttock circ? 

A5) We thank the reviewer for pointing out the unspecified conversion between the measurements. The conversion from A-O to ANSUR88 body measurements is as follows: 

(Our label – ANSUR88 attribute name) 

A (head circumference) - headcircumference 

B (neck circumference) - neckcircumference 

C (shoulder-to-crotch) - shoulder_to_crotch = sittingheight - (stature - acromialheight) 

D (chest circumference) - chestcircumference 

E (waist circumference) - waistcircumference 

F (hip circumference) - buttockcircumference 

G (wrist circumference) - wristcircumference 

H (bicep circumference) - bicepcircumferenceflexed 

I (forearm circumference) - forearmcircumferenceflexed 

J (arm length) - arm_length = acromialheight – wristheight 

K (inside leg length) - inside_leg_length = crotchheight - lateralmalleolusheight 

L (thigh circumference) - thighcircumference 

M (calf circumference) - calfcircumference 

N (ankle circumference) - anklecircumference 

O (shoulder breadth) - biacromialbreadth 

Regarding height and weight, height = stature * 0.001, and weight = weightkg * 0.1. The measurement conversion specification is now added to the updated manuscript (Table 4). 

R6) I have not been able to duplicate the mean absolute errors using regressions (with your four “interaction” terms) in the ANSUR88 dataset. The MAE for these regressions (male only) are (mm): 

HEAD_CIRC 10.5 

NECK_CIRC.BASE 11. 

CHEST_CIRC 24.1 

WAIST_CIRC.OMPHALION 33.1 

(Note that I had to guess which dimensions you had in mind since you do not specify which ANSUR or CAESAR dimension you used.)  

A6) We thank the reviewer for his/her effort to reproduce our results and we apologize for not specifying the exact ANSUR attributes. We have now updated our manuscript with the specifications about the ANSUR measurements. Please note that your obtained values very much correspond to the reported values in Table 5 (ANSUR, measures A, B, D, E, previously Table 4), but are not exactly the same probably due to differences in the train/test set splits.  

R7) Comparing to Table 4, which I assume lists MAE (the table caption does not specify the metric), I see very large discrepancies. More generally, based on my familiarity with anthropometric data, including a large amount of experience with imputing dimensions from other dimensions, I find your error values to be implausibly small, if in fact these are MAE. Please help me to understand how you achieved these results. 

A7) The reviewer is right: the error values in Table 5 (previously Table 4) are implausibly small, to be actually found in reality. But please read Table 5 and the related text once again. Table 5 represents both for our method and competitive methods noise-free scenarios, that is, using “ideal” (ground truth) input data and settings. For completeness, we wanted to show the performance of our and other methods in such idealistic scenarios (note that under such superb conditions some other methods show even more implausible results than ours). Therefore, we have never claimed that these (Table 5) are realistic results, one will typically encounter in practice.  

R8) Showing the performance of the regressions within ANSUR and/or CAESAR standard dimensions alone would go a long way to helping to clarify what you’re doing. [Note your reference to Gordon et al. has the authors and title of ANSUR II (2012) but a date of the publication of ANSUR88 (i.e., 1989). Which dataset did you use? It won’t affect the results much, but you should be clear, particularly because the definitions of a few measurements changed.] 

A8) We have also now corrected specification about ANSUR or CAESAR dataset. 

R9) Your presentation of the regressions is inadequate, particularly since this is the focus of the paper. Please present the full regression models (all coefficients for I=4 and I=2) along with p values, adjusted R^2 values, MAE (since that’s your preferred error metric) and root-mean-square errors for regressions with male & female data. 

A9) We agree with the reviewer that the presentation of our regression model should have been more complete. To confirm the necessary assumptions of the linear regression model, we now add and comment on the residuals, p-values, adjusted R^2 values, MAEs, and RMSEs (Section 5.1). 

R10) Please present additional discussion, with references, of the validity and likely errors in self-reported stature and weight. Modeling with a random component (uncorrelated?) would be expected to inflate your errors, but as noted above they are implausibly small as-is. 

A10) Note that the results presented in Table 6 (previously Table 5) are realistic. We agree that it would be more accurate to use more complex self-reporting model(s), i.e. the distributions that are correlated with other variables, than simply adding uncorrelated normal distributions with mean=0. For example, underweight people tend to overestimate their weight and vice versa (as shown in Cawley et al., “Reporting error in weight and its implications for bias in economic models” on NHANES data). Therefore, our model will probably wrongly estimate body measurements of the more extreme samples. However, we never claim that our model outperforms previous results for all examples, instead, we propose a simple and straightforward baseline that works well on average, and show that the “average” is comparable to the state-of-the-art in 2D- and 3D-based body measurement estimation. 

R11) Note that scan-extracted stature is not equivalent to manually measured stature. Values tend to be inflated by hair artifacts and reduced by a scan posture that is different from the maximally erect posture used to record stature. Please include this fact and the likely effects on your analysis in your presentation of the actual dimensions you used for each study (particularly when you normalized silhouette- and scan-based data to “ground truth” stature) and differences between scan-extracted and manual measurements. 

A11) As previously stated, all the samples are expected to be fully erect such as the ones in the training data. This simplifies body measurement, as well as assuring that the stature is not affected by changing posture. Of course, we can’t expect that each person will always take a perfectly erect pose, e.g., that the posture will not be affected by breathing. We now clearly point out all our assumptions, such the fully erect pose, hard artifacts, tight clothes, and others, in Section 5.5. 

R12) Please incorporate a more meaningful discussion (at least a paragraph) of allowable error. Gordon et al. (2012) has an excellent discussion, with references and procedures, in Section 7. Since you’re comparing results to these values, it’s critical that you help the reader to understand what these “expert errors” actually mean. 

A12) Please note that the allowable or the expert error is already described in the original manuscript as part of Related Work (Traditional Anthropometry), Section 2. According to the reviewer’s suggestion, we now add more details about the allowable error in Section 5.5. (Limitations, Assumptions, and Future Guidelines). 

Round 2

Reviewer 2 Report

Thanks for the outstanding and _very_ fast response to my review. I hope to see this article published soon.

My #1 complaint is that you are hiding your conclusion! Your abstract proposes a new “baseline” but doesn’t actually come out and say that (1) you evaluated the actual performance of state of the art methods for estimating body dimensions and (2) you found they worked worse, in most cases, than just asking people for their height and weight and doing some linear predictions. You say you “achieve strong performance on public datasets” but you achieve performance similar to or exceeding the (self-proclaimed) state-of-the-art methods. I don’t know if you’re just trying to avoid provoking potential reviewers, but at this point I hope you’re willing to be more direct in the abstract (which will be read by probably 100X more people than read the whole article). Buried in the article you write:  “[a] simple and straightforward baseline that works well on average, and show that the “average” is comparable to the state-of-the-art in 2D- and 3D-based body measurement estimation.” Put that (or an even more strong and succinct statement) in the abstract!

R3 – The key thing to recognize with the hair artifact is that it affects female scans more than male, even in military data. CAESAR is particularly bad in that regard. I certainly was not suggesting to discard the data, but just to note that measuring to the top of the scan is not the best way to estimate stature from a scan. Instead, using a regression model based on eye height or tragion height is more robust. (But beware the variability in head orientation!) I’m not suggesting to change the paper, that’s just for your future consideration in your work.

R4 – “…all the samples are expected to be fully erect…” As far as I know, none of the scans you’re referencing are taken in a fully erect posture, i.e., the posture in which stature is measured. This is yet another tricky thing with scans. The “A” pose used in CAESAR, ANSUR II, and other studies is something less than erect (comfortable), may not have the Frankfort plane horizontal, and the feet are spread more than in the standard anthropometric posture. (The reason stature is measured maximally erect rather than in comfortable standing is that maximally erect has the best reproducibility, not because it’s relevant for any applied purpose.) Hence, we expect the top of the (bald) head to be somewhat lower in scans than stature. We’ve found that this difference (between A-scan “stature” and stature) to be quite variable and associated with age. But, for your purposes, this means that your predicted (from self-report) stature is probably greater than the scan height (neglecting hair artifacts).

R7 – Do you mean Table 6? My values are similar to the bottom row of Table 6, but the Table 5 values are considerably lower. I think I understand what you are getting at here (these unrealistically low values based on an assumption used to explore the sources of discrepancy) but if I did not understand on first reading I think you should assume that even some of the more knowledgeable among your readers may also not understand how you derived the values, so please add a few more words of explanation, ideally in the caption of Tables 5 and 6. Or at least highlight the key words that differ between the captions: deterministic variables vs. stochastic.

R9 – Thanks for this more-complete presentation, which very substantially increases the value of the paper to people working in this area. (I did not check your tables, so I’m going to assume they are accurate!)

R10 – I take your point here. Your results are very convincing even without considering self-report bias.

Additional comments:

Thanks for including the details regarding the regressions. I would not have challenged you on homoscedasticity etc. (although the key to homoscedasticity is that the variance remains the same across the range of the independent variable). Research in the last 10-15 years has also shown the robustness of the linear model to violations of, e.g., normality, but I appreciate that you examined those things. However, I wouldn’t conduct a regression with the ensemble of variables you specified, because of the problem of collinearity of the predictors. That is, your coefficients are unstable because of the high correlation among the predictors (stature vs body weight, body weight vs BMI). In my own work, I would not include both stature and body weight in a regression, because the coefficients become uninterpretable and small changes in the sample result in relatively large changes in the coefficients due to the correlation between the predictors. BMI was invented to provide a measure of body weight less correlated with stature, so a better regression would use stature, BMI, and perhaps those squared terms. But I don’t think this is a substantial limitation, because I think readers would be best served by understanding your results and creating appropriate prediction models of their own. (And if they use your models directly, they won’t go wrong as long as they don’t try to interpret the coefficients.) Nonetheless, you might consider updating the results, at least for your own purposes, to substitute BMI for weight rather than including both.

I think the paper would be better without Sections 5.2 and 5.3. Many other papers address the effects of the PCs on (qualitative) body shape, and I think it distracts from your central point. Similarly, 5.3 presents some exploration related to a question (difference between the datasets) that I think would be better addressed by just a sentence or two. Figure 7 is colorful, distracting, and difficult to interpret, so I suggest you delete it.

On the whole, I’m very impressed with this work. Thanks for bringing this critical information to the community. But PLEASE state the conclusion strongly in the abstract! Otherwise I fear few people will ever see it. "Baseline" in the title doesn't sound very exciting. Maybe a new title? "Self-reported stature and body weight predict body dimensions as well as state-of-the-art deep-learning methods?" THAT will get some citations! 

Author Response

We once again thank the reviewer for relevant and detailed suggestions. We thoroughly considered each point and now offer our final, revised version of the manuscript.

Most notably, encouraged by the reviewer, we updated our title to "Linear Regression vs. Deep Learning: A Simple Yet Effective Baseline for Human Body Measurement". We also further strengthened our abstract and conclusion w.r.t. the major results.

We added brief remarks to Section 5.5 on how does fully-erect vs. A-pose affect stature measurement, and that hair artifacts particularly affect female subjects.

We made the difference between Tables 5 and 6 more clear by editing the captions, as suggested by the reviewer.

Finally, we used the title proposed by the reviewer, "Self-reported stature and body weight predict body dimensions as well as state-of-the-art deep-learning methods?" to write one more powerful sentence in our new conclusion.

This manuscript is a resubmission of an earlier submission. The following is a list of the peer review reports and author responses from that submission.

Round 1

Reviewer 1 Report

This work proposes to simplify human body measurement by using a linear regression model that takes height and weight of as inputs, and the results can somehow show the effectiveness of the method.

However, the method itself is now presented clearly. The current linear model is shown as  y=ax+b. How is the linear model regressed? It looks much too simple in the current format. The regression details are not presented at all.

Besides, the writing is not well checked with much errors, such as "A very popular research topic is pose and shape and estimation ..." and "we experiment with difffferent levels of added noise to input..." ...

Author Response

We thank the reviewer for the comments on our manuscript.

Q1: However, the method itself is now presented clearly. The current linear model is shown as  y=ax+b. How is the linear model regressed? It looks much too simple in the current format. The regression details are not presented at all.

A1: We have updated the linear regression model definition. More precisely, we have added two additional equations which describe the regression model and its standard analytical solution, and have extended the text accordingly (lines 145-157).

Q2: Besides, the writing is not well checked with much errors, such as "A very popular research topic is pose and shape and estimation ..." and "we experiment with difffferent levels of added noise to input..." ...

A2: Regarding the indicated spelling and grammar errors (lines 41 and 143) we have corrected them. Additionally, we have also carefully re-read the manuscript and have further improved the English language and the clarity of the exposition.

Reviewer 2 Report

This is a research article. A simple linear regression method is used to estimate the human body measurement/parameters. Writing English is a mature level, therefore there is no need for language editing. Literature review is enough to be comprehensive. Some minor questions are arised.

- Do you consider any method to estimate the precision of your computed volume? There exist some precision estimation methods for volumetric values.

- Have you consider to use the surface matching as a quantitative evaluation method? Volumetric discrepancies can also be gauge in this way.

Author Response

We thank the reviewer for the comments and an excellent evaluation. We answer the two questions point-by-point:

Q1: Do you consider any method to estimate the precision of your computed volume? There exist some precision estimation methods for volumetric values.

A1: To the best of our efforts, we did not find a way to objectively evaluate the computed volume. As mentioned in the original manuscript, we use the standard algorithm for estimating the volume from the 3D mesh [63] as implemented in the trimesh package [72]. However, the problem is that we do not have the ground-truth value for the volume. An approximative evaluation would be possible if the ground-truth human body weight is known; then we would be able to use a reverse formula [32, 64] for calculating the volume from the weight, however, ground-truth weights are also not available.

Q2: Have you consider to use the surface matching as a quantitative evaluation method? Volumetric discrepancies can also be gauge in this way.

A2: We have added Table 7 on page 10 for the comparison to the state-of-the-art with respect to the mean per-vertex error (MPVE). The text describing Table 7, the PVE metric, and the novel MPVE metric, is highlighted on page 9 (lines 266-276).

Reviewer 3 Report

The work presented for review is very interesting because of: 1. Topic 2. methodology 3. The way to solve the problem 4 Research results 5 Discussion and conclusions

Therefore, it can be concluded that in all areas of research the validity and scientific nature of the problem being solved are maintained. So what are the remarks that arise after reading the work: 1. As the authors themselves state, the problem is not widely described in the literature. The work, therefore, can become the leading study in this field. If we approach it this way, I am looking for some universal guidelines for future research in my work. Technical and methodological guidance. In the work, these contents are, however, required to be clearly marked. There is an editorial note.

  1. I was looking for a certain research universalism in my work. However, I had the impression that the research results were related to the problem in the clothing industry. Here, I would ask the authors to answer whether my perception is correct or incorrect. If wrong, they would point to the content that is more universal.
  2. I believe that in this type of work special attention should be paid to the level of measurement error and the reasons for its occurrence. In my opinion, it is also found in many workplaces, but for ease of reception it would be necessary to collect and analyze this problem in one place. This is an editorial note.

This is an editorial note. I rate the work highly in terms of content and my comments as a reviewer relate to the editorial sphere.

Author Response

We thank the reviewer for detailed comments. We answer the questions point-by-point:

Q1: As the authors themselves state, the problem is not widely described in the literature. The work, therefore, can become the leading study in this field. If we approach it this way, I am looking for some universal guidelines for future research in my work. Technical and methodological guidance. In the work, these contents are, however, required to be clearly marked. There is an editorial note.

A1: We are very glad that the reviewer considers our paper as a potential future leading study. Regarding this consideration, we extend our discussion in Subsection 5.5. Limitations and Future Guidelines (page 14), updating and adding lines 376-394. In the conclusion, we point out that the proposed linear regression should be used as a baseline for future evaluations. Also note that one of the contributions of our paper is the definition of the standard body measurement extraction protocol for template 3D meshes, which is an important future guideline for the method evaluation.

Q2: I was looking for a certain research universalism in my work. However, I had the impression that the research results were related to the problem in the clothing industry. Here, I would ask the authors to answer whether my perception is correct or incorrect. If wrong, they would point to the content that is more universal.

A2: Please note that our approach is not limited to the clothing industry and virtual try-on. We believe that the approach, the results, and the guidelines presented in our work can also be used for many other anthropometric applications, especially ergonomics and entertainment. For example, knowing human body dimensions can help in designing the ergonomic workplace - chairs, tables, and proper working device positioning. In entertainment, it can be used to create more accurate avatar shape for VR or AR. The results can also have an impact on fitness and semi-automatic healthcare. For example, the body shape changes can be tracked over time, which indicates and visually demonstrates the progress over time.

Q3: I believe that in this type of work special attention should be paid to the level of measurement error and the reasons for its occurrence. In my opinion, it is also found in many workplaces, but for ease of reception, it would be necessary to collect and analyze this problem in one place. This is an editorial note.

A3: In the original manuscript, we dedicate special attention to the measurement (estimation) errors. In particular, we provide:

- detailed analysis of the standard deviation and maximum error values in Table 6,

- qualitative analysis of the per-vertex errors in Figure 4,

- an analysis of body measurement errors with respect to added height and weight noise in Figures 7 and 8,

- comments on the sources of errors occurring in the previous works (Subsection 5.3. On Image-Based Mesh Regression),

- more detailed comparison between the baseline models, as well as the overall comparison to more complex models (Subsection 5.4. More Complex Models (Augmented Baseline)).

We dedicate most of Subsection 5.5. Limitations and Future Work to discuss the possible reasons for body measurement errors.

In the revised version, we further comment on the mean and standard deviation values with respect to body measurements of different absolute sizes (lines 258-260). We also extend our discussion on the possible reasons for body measurement errors, while providing future guidelines to mitigate these errors (lines 385-394).

Overall, we now believe that we properly addressed the problem of measurement errors and the possible reasons for their occurrence.

ADD: Additionally, we use the response to reviewer 3 to point out that we have updated the second part of Table 8 (linear regression coefficients for females), which we, unfortunately, omitted in the original version.